# Altered Blood–Brain Barrier Dynamics in the C9orf72 Hexanucleotide Repeat Expansion Mouse Model of Amyotrophic Lateral Sclerosis

**DOI:** 10.3390/pharmaceutics14122803

**Published:** 2022-12-14

**Authors:** Yijun Pan, Yoshiteru Kagawa, Jiaqi Sun, Bradley J. Turner, Cheng Huang, Anup D. Shah, Ralf B. Schittenhelm, Joseph A. Nicolazzo

**Affiliations:** 1Drug Delivery, Disposition and Dynamics, Monash Institute of Pharmaceutical Sciences, Monash University, 399 Royal Parade, Parkville, VIC 3052, Australia; 2Department of Organ Anatomy, Tohoku University Graduate School of Medicine, 2-1 Seiryomachi, Aobaku, Sendai 980-0872, Miyagi, Japan; 3Florey Institute of Neuroscience and Mental Health, University of Melbourne, Parkville, VIC 3052, Australia; 4Perron Institute for Neurological and Translational Science, Queen Elizabeth Medical Centre, Nedlands, WA 6009, Australia; 5Monash Proteomics & Metabolomics Facility, Biomedicine Discovery Institute, Monash University, Clayton, VIC 3800, Australia; 6Monash Bioinformatics Platform, Biomedicine Discovery Institute, Monash University, Clayton, VIC 3800, Australia

**Keywords:** amyotrophic lateral sclerosis, blood–brain barrier, brain microvascular endothelial cells, C9orf72 gene, proteomics

## Abstract

For peripherally administered drugs to reach the central nervous system (CNS) and treat amyotrophic lateral sclerosis (ALS), they must cross the blood–brain barrier (BBB). As mounting evidence suggests that the ultrastructure of the BBB is altered in individuals with ALS and in animal models of ALS (e.g., SOD1^G93A^ mice), we characterized BBB transporter expression and function in transgenic C9orf72 BAC (C9-BAC) mice expressing a hexanucleotide repeat expansion, the most common genetic cause of ALS. Using an in situ transcardiac brain perfusion technique, we identified a 1.4-fold increase in ^3^H-2-deoxy-D-glucose transport across the BBB in C9-BAC transgenic (C9) mice, relative to wild-type (WT) mice, which was associated with a 1.3-fold increase in brain microvascular glucose transporter 1 expression, while other general BBB permeability processes (passive diffusion, efflux transporter function) remained unaffected. We also performed proteomic analysis on isolated brain microvascular endothelial cells, in which we noted a mild (14.3%) reduction in zonula occludens-1 abundance in C9 relative to WT mice. Functional enrichment analysis highlighted trends in changes to various BBB transporters and cellular metabolism. To our knowledge, this is the first study to demonstrate altered BBB function in a C9orf72 repeat expansion model of ALS, which has implications on how therapeutics may access the brain in this mouse model.

## 1. Introduction

Amyotrophic lateral sclerosis (ALS) is a fatal neurodegenerative disease characterized by selective damage to motor neurons in the brain and spinal cord. The pathogenesis of ALS remains largely unknown, although there are a wide range of potential mechanisms related to neurodegeneration, including protein aggregation, altered RNA processing, oxidative stress, excitotoxicity and neuroinflammation [1,2,3]. Individuals with ALS present with progressive voluntary muscle weakness, paralysis and atrophy, followed by respiratory complications and failure [4]. ALS typically occurs between the age of 40 and 70, with an incidence rate of 2 in 100,000 people and a prevalence rate of 4–7 cases per 100,000 people [5]. Due to marked heterogeneity in its clinical phenotypes and aggressive progression of the disease, therapeutic treatments are limited, and patients usually survive for 3–5 years after symptom onset [6]. The genetic component of the pathogenesis in 70% of individuals with familial ALS has been identified; mutations in chromosome 9 open reading frame 72 (*C9orf72*) (40%), superoxide dismutase 1 (*SOD1*) (20%), fused in sarcoma (*FUS*) (5%), and TAR DNA-binding protein 43 (*TARDBP* encoding TDP-43) (5%) [2]. The repeat expansion in C9orf72 has also been identified in 5–10% of individuals with sporadic ALS [3], making C9orf72 mutations the most commonly known genetic cause of ALS. C9orf72 mutations are also a frequent cause of frontotemporal dementia (FTD) which underscores the genetic and pathological overlap with ALS [2]. 

Like other neurodegenerative diseases, such as Alzheimer’s disease and Parkinson’s disease, altered blood–brain barrier (BBB) and blood-spinal cord barrier (BSCB) dynamics have been reported in individuals with ALS and mouse models [7]. These barriers are composed of a lining of brain microvascular endothelial cells (BMECs) that serve to precisely regulate the transport of substances (including drugs) between the blood and the brain/spinal cord. In individuals with ALS and in SOD1^G93A^ mice [8], reduced expression (at mRNA or protein levels) of tight junction proteins has been reported [9,10], and increased paracellular permeability has also been noted [8,11]. Increased expression/function of efflux transporters (P-glycoprotein, P-gp and breast cancer resistance protein, Bcrp) has also been reported in individuals with ALS and both SOD1^G93A^ and SOD1^G86R^ mice [12,13]. To date, most animal studies have been performed in SOD1-related ALS mouse models, which only account for 20% of familial ALS cases. While C9orf72 repeat expansions are the most common (40%) genetic link in familial ALS and 5–10% of sporadic ALS, BBB dynamics in a mouse model with the C9orf72 mutation has yet to be assessed.

The C9orf72 gene contains 11 exons and healthy individuals commonly have between 2 and 23 GGGGCC (G_4_C_2_ hexanucleotide) repeats in the first intron, while individuals with ALS usually have hundreds to thousands of repeats [14,15]. Abnormal hexanucleotide repeat expansions are proposed to cause ALS pathology via three non-exclusive mechanisms. The first mechanism is through a gain of toxicity due to bidirectional transcription of C9orf72 repeat expansions, which leads to the formation of RNA foci, resulting in sequestration of RNA binding proteins and dysregulation of RNA splicing, transport and translation [16,17]. The second mechanism involves aggregation of dipeptide repeats via repeat-associated non-ATG translation which contribute to the formation of inclusions positive for TDP-43 protein, resulting in the disruption of mRNA splicing [18]. The third mechanism is believed to be a partial loss of function of C9orf72, which has been shown to influence lysosomal trafficking and autophagy processes, the alteration of which leads to aggregation of pathological proteins including TDP-43 [19]. Overall, misfolding and mislocalization of proteins due to pathological repeat expansions in C9orf72 disrupt multiple intracellular functions such as RNA metabolism, protein homeostasis, and mitochondrial function, possibly resulting in ALS pathophysiology [20]. An ALS mouse model based on C9orf72 repeat expansion (FVB-C9orf72 BAC mouse; C9-BAC) was successfully generated in 2016, with the transgenic mice (C9) exhibiting decreased survival, hind limb paralysis, muscle denervation, and motor neuron loss that is relevant to ALS, especially in females [21]. In addition, females also revealed signs of impaired cognition and anxiety indicative of FTD [11]. While the status of the BBB in other mouse models of ALS have been reported, characterization of the BBB in the C9-BAC model is a critical knowledge gap to be bridged, and may assist to unravel approaches that could be used to improve central nervous system (CNS) drug delivery in ALS.

In this study, we characterized the dynamics of the BBB in the C9-BAC mouse model at an early symptomatic age. This age (i.e., 145–150 days old) was selected as we were interested in potential BBB alterations at an age where significant pathology is observed, albeit severe motor dysfunction has not yet occurred. We evaluated the function of the BBB by assessing the transport of common probes, e.g., ^3^H-diazepam and ^14^C-caffeine for passive transcellular diffusion and ^14^C-sucrose for paracellular diffusion and determination of brain vascular volume. ^3^H-digoxin was included as a P-gp substrate as increased P-gp activity has been reported in other rodent models of ALS [7]. ^3^H-oleic acid was included to assess fatty acid transport processes given that oleic acid is one of the most abundant fatty acids in the brain. The BBB transport of ^3^H-2-deoxyglucose (^3^H-2DG) was included to assess the function of glucose transporter 1 (Glut1), which is important as altered glucose metabolism has been reported in other ALS mouse models and individuals with ALS [22]. Finally, ^14^C-(L)-alanine was selected as a probe to assess amino acid trafficking processes, because altered amino acid levels have been reported in the plasma and cerebrospinal fluid of individuals with ALS [23]. Proteomic analysis was also performed on brain microvascular endothelial cells (BMECs) isolated from wildtype (WT) and C9 mice to further assess any alterations to the proteome of the BBB in this model. We specifically compared the relative abundance of important tight junction and gap junction proteins, and key adenosine triphosphate (ATP)-binding cassette (ABC) transporters and solute carrier (SLC) transporters between WT and C9 mouse BMECs. Differential expression and functional enrichment analysis were also performed to explore particular proteins and cellular processes/pathways modified in the C9 BMECs. To our knowledge, this is the first study to characterize the BBB of a C9orf72 repeat expansion mouse model, both at a functional and proteomic level, providing insight into potential approaches that can be explored to improve CNS drug delivery in this mouse model, and ultimately, individuals with ALS. 

## 2. Materials and Methods

### 2.1. Materials

All radioactive probes (^3^H-diazepam, ^14^C-caffeine, ^14^C-sucrose, ^3^H-digoxin, ^3^H-oleic acid, ^3^H-2DG, and ^14^C-(L)-alanine) were purchased from American Radiolabeled Chemicals (Saint Louis, MO, USA). Solvable and Ultima Gold scintillation cocktail were obtained from Perkin Elmer (Waltham, MA, USA). Pierce™ BCA Protein Assay Kit, sequencing grade trypsin, TMTpro™ 16plex Label Reagent Set and other reagents used for proteomics were purchased from Thermo Fisher Scientific (Waltham, MA, USA). A mouse Glut1 ELISA kit was sourced from Wuxi Donglin Sci & Tech Development Co., Ltd. (Wuxi, China). Dulbecco’s Modified Eagle Medium (DMEM), bovine serum albumin (BSA) and phosphate-buffered saline (PBS) were purchased from Sigma-Aldrich (St. Louis, MO, USA).

### 2.2. Animals

Animal experiments were approved by the Monash Institute of Pharmaceutical Sciences Animal Ethics Committee (MIPS.21542) and performed in accordance with the National Health and Medical Research Council guidelines for the care and use of animals for scientific purposes. Transgenic C9-BAC (FVB/NJ-Tg(C9orf72)500Lpwr/J, strain number 029099) mice were purchased from the Jackson Laboratory (Bar Harbor, ME, USA) and maintained on FVB/NJ background. Female Tg(C9orf72)500Lpwr (C9) mice and their wildtype (WT) littermates were bred at Monash Animal Research Platform (Parkville, VIC, Australia) and the genotype of the mice was confirmed using Transnetyx^®^ Genotyping services (Cordova, TN, USA). C9 mice were generated by Liu et al. [21], who reported that hemizygous C9 mice appear normal with no overt cage behavior abnormalities up to 16 weeks of age. However, an acute, rapidly progressive disease (inactivity, labored breathing, sudden weight loss, hindlimb weakness, motor neuron degeneration/disease throughout the motor unit, paralysis and death) was reported in ~30–35% of C9 females between 20–40 weeks. As we were interested in the impact of this genetic mutation on the BBB at an early symptomatic age, female mice at 145–150 days old were used in this study. Strain matched non-transgenic littermates were used as WT controls. Therefore, WT and C9 female mice were housed with ad libitum access to standard rodent chow and water until 145–150 days of age for end point experiments.

### 2.3. In Situ Transcardiac Brain Perfusion to Assess BBB Transport 

The transport of probe drugs across the BBB was assessed using an in situ transcardiac brain perfusion technique as previously described [24]. Surgical anesthesia of mice (*n* = 4–8 per genotype) was established using ketamine (133 mg/kg) and xylazine (10 mg/kg), and the mice were kept on a warm surgery pad throughout anesthesia. The thoracic cavity of the mice was cut open to expose the heart, the descending aorta was clamped to restrict blood flow to the peripheral organs, and both jugular veins were severed. Immediately thereafter, warm (37 °C) Krebs-ringer bicarbonate buffer containing the radioactive probe (0.05 µCi/mL) was injected into the left ventricle of the heart and maintained at a rate of 10 mL/min for 1 min controlled by a Harvard infusion pump (Harvard Apparatus, Holliston, MA, USA). After 1 min, the perfusion was stopped, and the brain was harvested and digested in 2 mL of Solvable™ at 70 °C for 6 h. A 200 µL aliquot of hydrogen peroxide (30% *v*/*v*) was added to neutralize the color. Ultima Gold scintillation cocktail (10 mL) was added to the brain sample and 2 mL of Ultima Gold scintillation cocktail was added to 100 µL of the radioactive perfusion fluid. Radioactivity in each sample was counted in a PerkinElmer 2800TR liquid scintillation analyzer (Waltham, MA, USA). The apparent brain distribution volume of the probe compound (brain:perfusate, mL/g) was calculated as previously described [24,25]. The vascular volume was quantified using brain:perfusate ratio of ^14^C-sucrose in C9 and their WT littermates, which was used for vascular volume subtraction for the calculation of apparent brain distribution volume of all other probes.

### 2.4. Brain Microvessel Enrichment for Glut1 Quantification 

Given results demonstrated that the BBB transport of ^3^H-2DG was increased in C9 mice, quantification of Glut1 levels in brain microvessels from WT and C9 mice was conducted. WT and C9 mice were humanely killed via cervical dislocation under anesthesia. Brains were immediately removed and 3–4 brains were pooled for subsequent processing, as previously described [25], to obtain sufficient microvessels for ELISA analysis of Glut1. In brief, mouse brains were homogenized using a Dounce homogenizer, and the microvessel enriched fraction (MEF) was separated from the homogenate by centrifugation using 17% (*w*/*v*) BSA in DMEM. The residual BSA was removed by rinsing the MEFs with ice-cold PBS and the total protein was extracted from the MEFs via 2 freeze–thaw cycles. A total volume of 200 µL of PBS was added to the MEFs and the samples were sonicated (2 × 10 s) prior to centrifugation. The total protein count for the supernatant was assessed using a Pierce™ BCA Protein Assay Kit by comparison to a standard curve developed using known BSA standard solutions. ELISA results were presented as Glut1 levels (ng)/mg total protein.

### 2.5. BMEC Isolation for Proteomic Analysis

To achieve a highly purified BMEC population (compared to that achieved via brain microvessel enrichment method described above), a magnetic-activated cell sorting (MACS) technique was employed. All procedures were performed according to manufacturer’s instructions (Miltenyi Biotec Inc., Bergisch Gladbach, Germany). WT and C9 mice were anesthetized using isoflurane and humanely killed via cervical dislocation. Immediately, the brain was removed, weighed, and digested using an Adult Brain Dissociation Kit (mouse and rat) (Miltenyi Biotec Inc.). From the pool of brain cells, endothelial cells were isolated using MACs mouse CD31 microbeads. The obtained BMECs were pelleted down and stored at −80 °C for proteomic analysis.

### 2.6. Proteomic Analysis of BMECs

BMEC samples were lysed in SDS lysis buffer (5% *w*/*v* sodium dodecyl sulphate, 100 mM HEPES, pH 8.1), heated at 95 °C for 10 min and then probe-sonicated before measuring protein concentration using a BCA kit. The lysed samples were denatured and alkylated by adding tris(2-carboxyethyl) phosphine hydrochloride and 2-chloroacetamide to a final concentration of 10 mM and 40 mM, respectively, and the mixture was incubated at 55 °C for 15 min. The proteins were trapped using S-Trap mini columns (Profiti, Farmingdale, NY, USA), and sequencing grade trypsin was added at an enzyme to protein ratio of 1:50 and incubated overnight at 37 °C. Tryptic peptides were sequentially eluted from the columns using (i) 50 mM triethylammonium bicarbonate, (ii) 0.2% *v*/*v* formic acid and (iii) 50% *v*/*v* acetonitrile, 0.2% *v*/*v* formic acid. The sequentially eluted fractions were pooled, concentrated in a vacuum concentrator and reconstituted in 40 µL of 200 mM HEPES, pH 8.5. Using a Pierce Quantitative Colorimetric Peptide Assay Kit (Thermo Fisher Scientific, Waltham, MA, USA), equal peptide amounts of each sample were labelled with the TMTpro 16plex reagent set (Thermo Fisher Scientific) according to the manufacturer’s instructions, considering a labelling strategy to minimize channel leakage. Individual samples were then pooled and high-pH RP-HPLC (1260 Infinity II, Agilent) was used to fractionate each pool into 12 fractions, which were acquired individually by LC-MS/MS to maximize the number of peptide and protein identifications.

Using a Dionex UltiMate 3000 RSLCnano system equipped with a Dionex UltiMate 3000 RS autosampler, an Acclaim PepMap RSLC analytical column (75 µm × 50 cm, nanoViper, C18, 2 µm, 100 Å; Thermo Scientific) and an Acclaim PepMap 100 trap column (100 µm × 2 cm, nanoViper, C18, 5 µm, 100 Å; Thermo Fisher Scientific, the tryptic peptides were separated by increasing concentrations of 80% *v*/*v* acetonitrile/0.1% *v*/*v* formic acid at a flow of 250 nL/min for 158 min and analyzed with an Orbitrap Fusion Tribrid mass spectrometer (ThermoFisher Scientific). The instrument was operated in data-dependent acquisition mode to automatically switch between full scan ms1 (in Orbitrap), ms2 (in ion trap) and ms3 (in Orbitrap) acquisition. Each survey full scan (380–1580 m/z) was acquired with a resolution of 120,000, an automatic gain control (AGC) target of 50%, and a maximum injection time of 50 msec. Dynamic exclusion was set to 60 sec after one occurrence. Keeping the cycle time fixed at 2.5 sec, the most intense multiply charged ions (z ≥ 2) were selected for ms2/ms3 analysis. Ms2 analysis used collision-induced dissociation (CID) fragmentation (fixed collision energy mode, 30% CID collision energy) with a maximum injection time of 150 msec, a “rapid” scan rate and an AGC target of 40%. Following the acquisition of each ms2 spectrum, an ms3 spectrum was acquired from multiple ms2 fragment ions using Synchronous Precursor Selection. The ms3 scan was acquired in the Orbitrap after higher energy collision dissociation with a resolution of 50,000 and a maximum injection time of 250 msec.

The raw data files were analyzed with Proteome Discoverer (Thermo Fisher Scientific) to obtain quantitative ms3 reporter ion intensities. Statistical analysis was performed using R statistical analysis software [26]. First, the dataset was filtered for high confident proteins. Next, the contaminant proteins were removed. The protein intensity data was converted to log2 scale and samples were grouped by conditions. The purity of the enriched BMEC samples was evaluated by assessing the abundance of protein markers for endothelial cells (von Willebrand factor, Vwf), astrocytes (Ndrg2), pericytes (platelet-derived growth factor receptor β; Pdgfr-β), and neurons (synaptophysin; Syp). It has been reported that the BBB integrity is compromised in individuals with ALS and mouse models of ALS, and therefore, we evaluated the brain vasculature proteome data for abundance changes in classical proteins implicated in BBB integrity [27], including zonula occludens-1 (Tjp1), zonula occludens-2 (Tjp2), claudin 5 (Cldn5), occludin (Ocln), cadherin 5 (Cdh5), junctional adhesion molecule 2 (Jam2), junctional adhesion molecule 3 (Jam3), F11 receptor (F11r) and endothelial cell adhesion molecule (Esam). As drug delivery is the focus of our study, the abundance of major ABC transporters and SLC transporters were compared between WT and C9 BMECs. In addition, protein-wise linear models combined with empirical Bayes statistics were used for the differential expression analyses. The limma package [28] from R Bioconductor was used to generate a list of differentially expressed proteins for each pair-wise comparison. A cutoff of the adjusted *p*-value of 0.05 (Benjamini-Hochberg method) along with a fold-change of 1.5 was applied to determine significantly regulated proteins in the pairwise comparison of WT vs. C9 BMECs. Functional enrichment analysis was performed using STRING online tools (https://string-db.org/, accessed on 10 July 2022).

### 2.7. Statistical Analysis of Data

All data are expressed as mean ± SEM. The comparisons between results obtained from WT and C9 mice were evaluated by Student’s unpaired *t* tests and two-way ANOVA with repeated measure where appropriate. Proteomics data analysis was detailed above and specified in the Results Section. Differences with a *p* value of less than 0.05 were considered statistically significant unless otherwise indicated in the text.

## 3. Results

### 3.1. BBB Transport of ^3^H-2DG Was Increased in C9 Mice Relative to WT Mice, in Association with Increased Brain Microvascular Glut1 Abundance

An in situ transcardiac brain perfusion technique was employed to characterize BBB transport processes in the C9-BAC mouse model. No significant difference in the cerebral vascular volume was noted between WT (0.025 ± 0.003 mL/g) and C9 (0.026 ± 0.003 mL/g) mice, using ^14^C-sucrose as a marker. The BBB transport of ^3^H-diazepam, ^14^C-caffeine, ^3^H-oleic acid, ^3^H-digoxin, and ^14^C-(L)-alanine was comparable between WT and C9 mice (Figure 1A–E), suggesting that passive diffusion, P-gp efflux, fatty acid uptake and neutral amino acid uptake was not affected in this mouse model. Interestingly, a 1.4-fold increase in the BBB transport of ^3^H-2DG was observed in C9 mice relative to WT mice (Figure 1F). To ascertain if this increase in the BBB transport of ^3^H-2DG was associated with increased expression of Glut1 at the BBB, brain MEFs were isolated and a 1.3-fold increase in Glut1 abundance was observed in brain MEFs isolated from C9 mice compared to WT mice (Figure 2).

### 3.2. The Tandem Mass Tag (TMT)-Proteomics Studies Revealed an Altered Proteome Profile of C9 BMECs Compared to WT BMECs 

Based on the results demonstrating altered BBB trafficking of ^3^H-2DG, a proteomics approach was undertaken to explore further alterations to the BBB in early symptomatic C9 mice compared to their WT littermates. Isolated BMECs from 8 biological replicates (two mouse brains were pooled for one biological replicate) were subjected to TMT-based quantitative proteomic analysis, and a total of 5267 proteins were quantified across all samples (Appendix A: Imputed normalized intensity). A relatively high abundance of the endothelial marker Vwf and low abundance of markers for astrocytes (Ndrg2), pericytes (Pdgfr-β) and neurons (Syp) were observed in the proteome profile, confirming the purity of the BMEC samples (Figure 3A). A main effect of markers (F_3,28_ = 111.8, *p* < 0.0001), but not of genotype, was revealed by a two-way ANOVA repeated measure.

Further characterization of the BBB was undertaken by comparing the abundance of Tjp1, Tjp2, Cldn5, Ocln, Cdh5, Jam2, Jam3, F11r and Esam (Figure 3B). A two-way ANOVA with repeated measures revealed an interaction effect between genotype and proteins of interest (F_8,63_ = 2.108, *p* = 0.048), and the post hoc Sidak’s multiple comparison test demonstrated a significant (14.3 ± 9.4%, adjusted *p* value = 0.0003) reduction in zonula occludens-1 abundance in C9 BMECs compared to WT BMECs. In addition, a total number of 18 ABC transporters and 83 SLC transporters were identified in BMECs. Table 1 summarizes the relative intensity of the most important membrane transporters for drug and endogenous molecule disposition [29]. Of note, a significant increase in Abc9, Eaat2, Asct2, Fatp1 and a significant decrease in Bsep and Taut were observed, which are important transporters for the disposition of endogenous molecules.

As per differential expression analysis, for all detected proteins, a total of 13 significantly differentially abundant proteins were identified, with 10 proteins exhibiting a >1.5 fold-change (Table 2). The volcano plot and heat map of significantly differentially abundant proteins are available in Appendix A. STRING was used to perform functional enrichment analysis for all identified proteins irrespective of whether they were significantly regulated. Here, we provided some examples of the Biological Process (BP), Molecular Function (MF), and Cellular Component (CC) relevant to BBB transport highlighted by the analysis (Figure 4). When comparing WT with C9 BMECs, the BP was enriched with positive regulation of monovalent inorganic cation transport (GO:0015672), regulation of exocytosis (GO:0017157), and cation transmembrane transport (GO:0098655); the MF was enriched with negative regulation of organic anion transmembrane transporter activity (GO:0008514), and inorganic molecular entity transmembrane transporter activity (GO:0015077), but positive regulation of monovalent inorganic cation transmembrane transporter activity (GO:0015077); and the CC was enriched with positive regulation of mitochondrial protein complex (GO:0098798), and collagen-containing extracellular matrix (GO:0062023). We also performed KEGG analysis, where extracellular matrix organization (MMU-1474244; positive regulation), SLC-mediated transmembrane transport (MMU-425407; both positive and negative regulation), the citric acid cycle and respiratory electron transport (MMU-1428517; positive regulation), oxidative phosphorylation (MMU-00190; positive regulation), and drug metabolism—cytochrome P450 (MMU-00982; negative regulation) were highlighted. The complete results of the functional enrichment analysis are available in Appendix A: Functional enrichment analysis.

## 4. Discussion

Altered BBB structure and function have been reported in SOD1-relevant rodent models of ALS and in individuals with ALS [7]. Given C9orf72 repeat expansions are the most common genetic cause of ALS, we characterized BBB transport function in the C9-BAC mouse model. The selection of probes was based on highlighting different BBB transport processes [24,30] that may be altered in this ALS model. We assessed the brain microvascular volume of these mice using ^14^C-sucrose, and no significant difference was noted between WT and C9 mice, suggesting that paracellular permeability across the BBB was not impacted in this particular mouse model. The brain microvascular volume in WT and C9 mice was similar to that previously reported in C57BL/6 mice (~0.02 mL/g) [24,31]. ^3^H-diazepam and ^14^C-caffeine were used to assess transcellular BBB diffusion of both a lipophilic and hydrophilic compound, respectively, and the brain:perfusate ratios obtained in the current study, which were not different between genotype, were comparable to the reported values in rodents for diazepam [24] and caffeine [32]. Overall, these data suggested that there is no brain vascular leakiness or altered BBB passive diffusion in C9 mice, relative to WT mice, using our established in situ transcardiac brain perfusion technique [24,33].

The above-mentioned observations are not in alignment with the impaired BBB integrity and increased vascular permeability reported in individuals with ALS and SOD1^G93A^ mice [8,11]. However, it has to be noted that human samples were collected from deceased individuals with ALS and therefore the samples represent pathology at terminal disease stage, which is different to our current study where we assessed BBB trafficking at an early symptomatic stage without significant overt disease progression. For animal studies showing altered vascular integrity, this observation was in mouse spinal cord microvasculature using electron microscopy and permeability assessed using Evans Blue, as opposed to brain microvascular tissue used in our studies. Brain microvascular tissue was also more relevant in our study given C9-BAC mice model both ALS and FTD. To make direct comparison between our results in the C9-BAC model with SOD1^G93A^ mice, in situ transcardiac brain perfusions should also be performed in SOD1^G93A^ mice in future studies. Regardless, the CNS delivery of drugs that utilize passive diffusion to permeate the BBB is not expected to be affected in the C9-BAC mouse model of ALS. Given the absence of functional changes in passive transcellular and paracellular permeability, we did not proceed to histological evaluation of the cerebral microvasculature (e.g., basement membrane thickness and vascular integrity).

A selective increase in microvascular expression of P-gp during disease progression in three ALS mouse models (SOD1^G93A^, SOD1^G86R^ and TDP-43^A315T^) has been reported in the cerebral cortex and spinal cord [12,13]. Increased P-gp expression was identified in the brain cortex and spinal cord homogenates of symptomatic mice in all three models [12], and a 1.5-fold increase in P-gp abundance was detected in microvessels isolated from the brains of pre-symptomatic SOD1^G86R^ mice compared to age-matched WT controls [13]. We assessed P-gp function initially, rather than expression, in C9-BAC mice at an early symptomatic age, using ^3^H-digoxin as a well-established substrate of P-gp [34], and no significant difference in BBB transport of ^3^H-digoxin was identified between WT and C9 mice. Given that there was no evidence of increased vascular leakiness as assessed using ^14^C-sucrose (which could theoretically mask any increased function of P-gp), we concluded that P-gp activity was not modified at the BBB in the C9-BAC model. The discrepancies between our results and those in other studies could be attributed to the different ALS pathology that the C9-BAC model represents relative to the SOD1-relevant models, and the different ALS disease stage of C9 mice assessed in our studies relative to those reporting alterations in symptomatic SOD1 mice. We also assessed the BBB trafficking of ^3^H-oleic acid and ^14^C-alanine, given that the levels of oleic acid in the spinal cord of individuals with ALS has been shown to be increased by 10% [35], and that the levels of alanine in the cerebrospinal fluid of individuals with ALS has been shown to be increased by 21% [23]. No significant difference in the BBB trafficking of ^3^H-oleic acid and ^14^C-alanine was observed between WT and C9 mice, suggesting that any increases in brain levels of oleic acid or cerebrospinal fluid levels of alanine reported in ALS are less likely due to changes in their transport across the BBB.

Interestingly, a significant increase in the BBB transport of ^3^H-2DG was noted in the C9 mice compared to their WT littermates. This observation did not appear to be in line with previous reports where glucose hypometabolism, as measured by PET, has been reported in spinal cords and several brain regions, including motor, frontal and occipital cortex of both individuals with ALS and in ALS animal models [36]. However, it has to be emphasized that our in situ transcardiac perfusion technique was performed over 1 min to evaluate the kinetics of ^3^H-2DG trafficking across the BBB, whereas the previously reported reduction in glucose homeostasis was assessed through measuring the incorporation of ^18^F-FDG into the brain parenchyma 40 min post-dosing via PET [36]. It has been proposed that glucose is transported across the BBB via Glut1 [37], whereas Glut3 is required to transport glucose from the brain interstitial fluid into neurons [38]. It is possible that the increased BBB transport of ^3^H-2DG that we observed reflects an increase in the availability of glucose uptake to compensate for the decreased glucose neuronal incorporation (as has been demonstrated via PET). Associated with the increased BBB trafficking of ^3^H-2DG was an increase in Glut1 abundance in brain MEFs isolated from C9 mice compared to WT mice. Further studies are required to explore the underlying mechanisms leading to this increased expression and function of Glut1 and to ascertain whether this may be exploited to enhance CNS delivery of therapeutics in ALS, for example by conjugating glucose-like entities to assist trafficking across the BBB [39].

We also employed proteomic analysis as an explorative approach to capture novel findings that might have been unidentified by our routine BBB functional characterization. Differentially expressed protein analysis and functional enrichment analysis were performed, which also provided insight into status of the BBB in the C9-BAC mouse model. In the current study, the purity of the BMEC samples was confirmed by the relative abundance of a marker for endothelial cells (vwf) relative to markers for astrocytes (Ndrg2), pericytes (Pdgfr-β) and neurons (syp) in the proteome profiles. While glial fibrillary acidic protein (Gfap), S100β and Ndrg2 are all recognized markers for astrocytes, Ndrg2 was selected due to its uniform expression in astrocytes (>40% of astrocytes have been found to be Gfap-negative), and its astrocyte-specific expression (S100β is also expressed in some mature oligodendrocytes, and choroid plexus epithelial cells) [40]. Pdgfr-β was selected for uniform and pericyte-specific expression [41] and Syp was employed as a neuronal marker, given it is one of the most commonly used markers of neurons [42]. A high abundance of vwf was noted in the proteomic analysis of our isolated BMECs, which was ~6 times higher than Ndrg2, and ~17 times higher than Pdgfr-β and Syp. Sharma et al. [43] have performed a proteomic analysis in different mouse brain regions, and assessed the relative abundance of these markers in brain tissues (without endothelial cell enrichment procedure). In brain homogenates, the abundance of vwf was found to be 44-, 65-, and 942-fold lower than Ndrg2, Pdgfr-β and Syp, respectively, suggesting that our samples are indeed highly enriched in BMECs.

Although we did not observe functional differences in tight junction function in C9 mice (through assessing ^14^C-sucrose transport across the BBB), we were still interested to assess the abundance of major tight junction and gap junction proteins [44] in BMECs that were isolated from the C9 and WT mice, as altered expression of tight junction proteins in the spinal cords (though not in the brain) have been reported in individuals with ALS and in the SOD1^G93A^ mouse model [45,46]. A mild (14.3 ± 9.4%) but significant reduction in the abundance of zonula occludens-1 was identified in BMECs isolated from C9 mice compared to those isolated from WT mice. This observation is in line with a previous report where a ~50% reduction in zonula occludens-1 was observed at the protein level in the spinal cord microvasculature isolated from SOD1^G93A^ mice relative to WT mice [45]. The less robust reduction in zonula occludens-1 abundance could be due to the different techniques used (i.e., untargeted proteomics in our studies vs. Western blotting in the SOD1^G93A^ mice), or the intrinsic difference between the models and the CNS regions assessed (i.e., we assessed BMECs from the cortex and not the spinal cord). Apart from zonula occludens-1, no significant changes in other tight junction and gap junction proteins were identified in our proteomics study. Regardless, this mild reduction in zonula occludens-1 abundance, as assessed using untargeted proteomics, did not result in any functional changes to paracellular permeability in the C9 mice, overall suggesting that the paracellular route is intact in the C9-BAC mouse model of ALS. 

We also specifically assessed the abundance of major ABC and SLC proteins in BMECs isolated from WT and C9 mice (Table 1). Some proteins were identified to be significantly modified in expression in C9 BMECs in this subset of proteomics data, however there are no clinically approved drugs that are known to interact with these transporters, and therefore such changes are less likely to affect CNS drug delivery in practice. Notably, no significant change to P-gp and Bcrp were identified in C9 BMECs compared to WT BMECs, while overexpression for P-gp and Bcrp has been observed in SOD1^G93A^ mice (brain cortex and spinal cord) and postmortem ALS human tissues (spinal cord) [7]. It is completely possible that the expression and function of P-gp and Bcrp are not yet altered in early symptomatic C9-BAC mice, and that such changes may occur with disease progression (as has been reported in SOD1^G93A^ mice albeit in spinal cord homogenates) [12]. In addition, different techniques used for quantitative analysis could also contribute to such discrepancies. Indeed, most other studies investigating the BBB in ALS mouse models or postmortem human tissues employed Western Blot or immunohistochemistry, rather than a high throughput proteomics technique. Despite this, we do not expect to reveal such differences in P-gp by undertaking these studies in the C9-BAC mice, given that we observed no change to P-gp function at the BBB using ^3^H-digoxin as a P-gp substrate. While a significant increase in Glut1 was observed in the brain MEFs from C9 mice relative to WT mice using an ELISA approach, our proteomics study did not reveal such changes in BMECs. These differences could be attributed to different isolation techniques. For the brain microvessel enrichment technique, there was no enzymatic digestion step, and therefore, there may be a greater amount of astrocyte end feet connections still attached to the microvasculature; while for the BMEC isolation by MACS and subsequent enzymatic digestion, we achieved samples with much lower astrocyte contamination. It is also possible that proteomics is less sensitive to detect mild changes (<1.5-fold) than ELISA [46]. Regardless, given that Glut1 function was increased at the BBB in C9 mice, as assessed by increased BBB trafficking of ^3^H-2DG, this transporter is still considered an exploitable target for improving CNS drug delivery in ALS.

Interestingly, alterations to the levels of some transporters, such as Eaat2 and Asct2, were identified, which are possibly important for disease pathology, albeit they may not be involved in drug transport across the BBB. Eaat2 is one of the most abundant subtypes of glutamate transporters in the CNS, playing a key role in maintaining glutamate concentration low in the extracellular space of the brain so as to avoid excitotoxicity [47]. Dysfunction of Eaat2 has been correlated with ALS [47]. It is possible that Eaat2 is upregulated at an early symptomatic stage in C9 mice as a compensatory mechanism to remove excessive glutamate, especially given that Eaat2 is expressed at the abluminal membrane of BMECs [48]. Similarly, being a glutamate transporter at the abluminal membrane of the brain microvasculature [49], the upregulation of Asct2 in C9 BMECs could also be part of the compensatory mechanism to avoid excitotoxicity from excess glutamate [50]. It was also observed that the abundance of Fatp1 (a transporter of oleic acid) [51] increased in C9 vs. WT BMECs. Though this observation is in agreement with the increased oleic acid levels observed in the postmortem spinal cord of individuals with ALS [35], it does not explain why BBB transport of ^3^H-oleic was not increased as assessed using a transcardiac in situ brain perfusion technique. This could be due to the fact that other proteins (such as fatty acid binding protein 5) [52] are essential in oleic acid uptake, and these proteins that did not appear to be affected in this study.

In line with the differentially expressed proteins which we noted in our study comparing WT and C9 BMECs, Hmgn3 and Nono have been shown to be differentially expressed in human ALS brains [53]. While they play roles in astrocyte function and possible glucose homeostasis [54,55,56]_,_ their role at the BBB and any impact on trafficking of entities into the CNS remains completely unknown. On the other hand, from the functional enrichment analysis, many GO terms and KEGG pathways that are relevant to transporter function and energy metabolism were highlighted; for example, cation transmembrane transport (GO:0098655), citric acid cycle and respiratory electron transport (MMU-1428517), and oxidative phosphorylation (MMU-00190). Future studies focusing on these changes at the ALS BBB may be considered; for example, the BBB transport of quetiapine, a cationic drug that is commonly used in individuals with ALS, can be assessed given the highlighted changes in cation transmembrane transport, and the metabolism of C9 BMECs can be investigated, given the highlighted changes in citric acid cycle and oxidative phosphorylation.

## 5. Conclusions

Identifying novel targets to overcome the restrictive nature of the BBB has the potential to improve CNS drug delivery in ALS. Using the C9-BAC mouse model of ALS, our studies have demonstrated that the BBB in the early symptomatic phase of pathology is mildly affected with a significant increase in Glut1 expression and function, and no observed alteration to paracellular diffusion, passive diffusion or P-gp mediated efflux. Our proteomics approach has identified other possible transporters that could be further investigated to provide a more detailed characterization of the BBB, which might highlight in addition to Glut1, other transporters that could be exploited for enhancing CNS drug delivery in ALS. Building upon our proteomics data, future studies are required to evaluate the BBB transport of different classes of drugs that possibly have modified access to the brain of mouse models of ALS; this will provide invaluable insight into how to design drugs for optimum brain uptake for this disease. Knowing this in advance can also lead to tailoring the dosages of non-ALS medicines to minimize their potential for brain toxicity. 

## Figures and Tables

**Figure 1 pharmaceutics-14-02803-f001:**
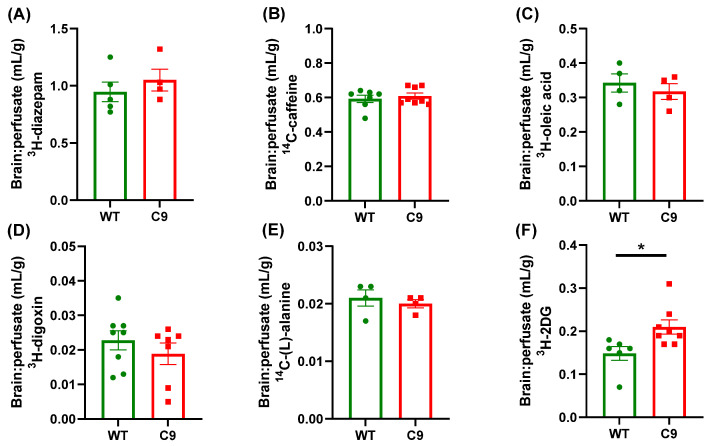
The brain-to-perfusate ratio of probe molecules in 145–150 day old female WT and C9 mice, with (**A**) ^3^H-diazepam to assess lipophilic passive diffusion, (**B**) ^14^C-caffeine to assess hydrophilic passive diffusion, (**C**) ^3^H-oleic acid to assess fatty acid transport processes, (**D**) ^3^H-digoxin to assess P-gp function, (**E**) ^14^C-(L)-alanine to assess the transport of a neutral amino acid, and (**F**) ^3^H-2DG to assess the function of Glut1. Data are presented as mean ± SEM (*n* = 4–8 mice per genotype), with * *p* < 0.05 using an unpaired Student’s *t*-test.

**Figure 2 pharmaceutics-14-02803-f002:**
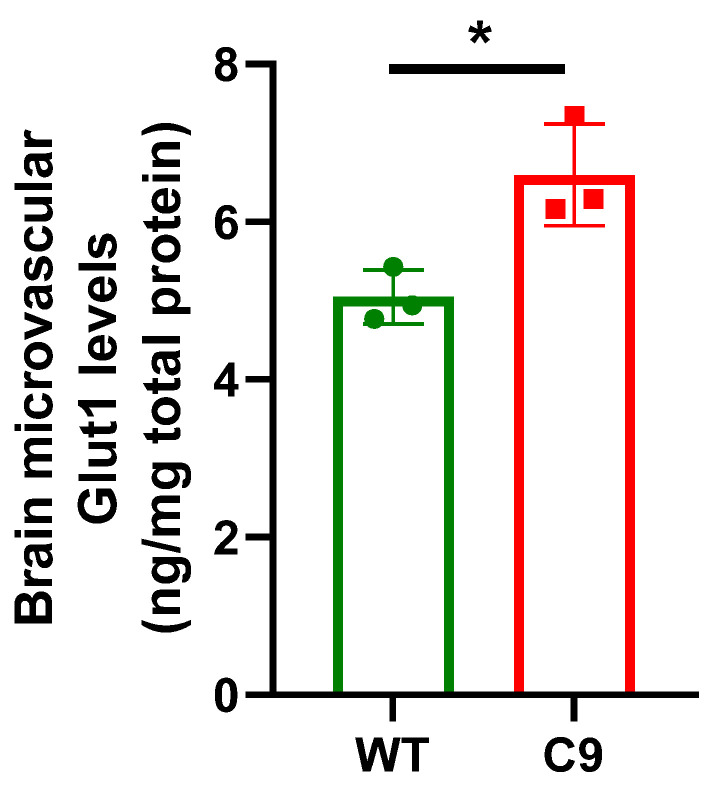
The abundance of Glut1 in brain MEFs from 145–150 day old female WT and C9 mice, with a significant 1.3-fold increase observed in brain MEFs from C9 mice compared to those from WT mice. Data are presented as mean ± SEM (*n* = 3 biological replicates per genotype; 3–4 mouse brains were pooled for each biological replicate), with * *p* < 0.05 using an unpaired Student’s *t*-test.

**Figure 3 pharmaceutics-14-02803-f003:**
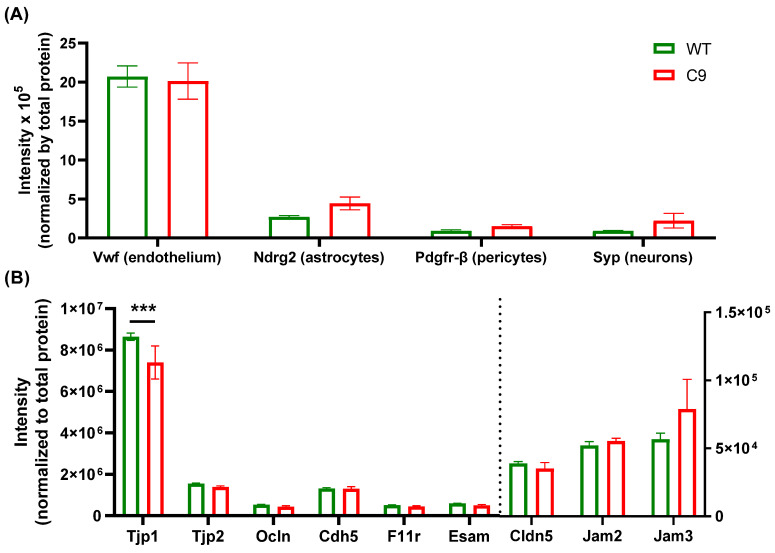
(**A**) The relative abundance of markers for endothelial cells (Vwf), astrocytes (Ndrg2), pericytes (Pdgfr-β) and neurons (Syp), and (**B**) microvascular integrity markers in BMECs isolated from 145–150 day old female WT and C9 mice, with the intensity of proteins on the left side of the dash line represented by the left Y axis and the intensity of proteins on the right side of the dash line represented by the right Y axis Data are presented as mean ± SEM (*n* = 8 mice per genotype), with *** *p* < 0.001 using a two-way ANOVA with post hoc Sidak’s multiple comparison.

**Figure 4 pharmaceutics-14-02803-f004:**
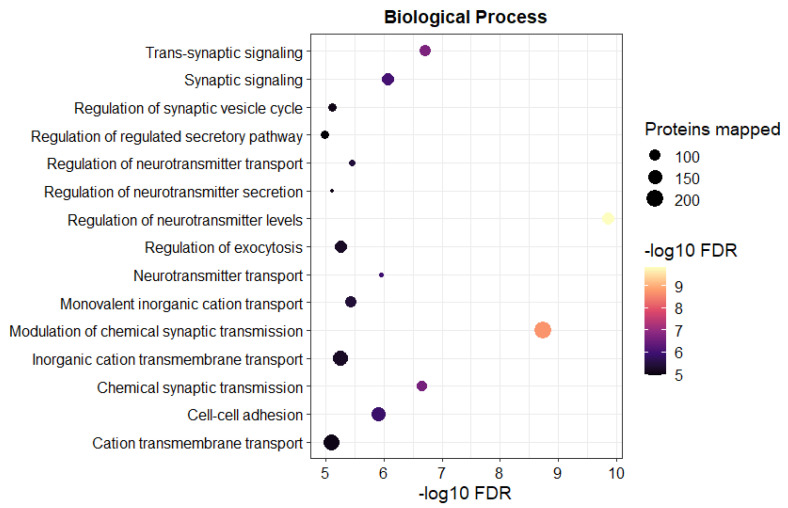
Representative results of functional enrichment analysis for BMECs isolated from 145–150 day old female WT and C9 mice (*n* = 8 per genotype).

**Table 1 pharmaceutics-14-02803-t001:** Relative abundance of important membrane transporters in BMECs isolated from 145–150 day old female WT and C9 mice, assessed using TMT-proteomics (* *p* < 0.05).

Gene	Protein Name	WT/C9 Fold-Change	*p* Value
ABC transporters			
Abca2	Abc2	1.18	0.12
Abca9	Abc9	0.66 *	0.03
Abcb1a	P-gp	1.23	0.14
Abcb4	Mdr3	1.09	0.20
Abcb11	Bsep	1.35 *	0.01
Abcc1	Mrp1	1.14	0.31
Abcc9	Mrp9	0.78	0.12
Abcg2	Bcrp	1.21	0.07
SLC transporters			
Slc1a2	Eaat2	0.42 *	0.05
Slc1a5	Asct2	0.77 *	0.04
Slc6a6	Taut	1.21 *	0.05
Slc2a1	Glut1	1.22	0.23
Slc7a5	Lat1	1.32	0.08
Slc16a1	Mct1	1.32	0.13
Slc22a8	Oat3	1.15	0.33
Slc27a1	Fatp1	0.70 *	0.03
Slc29a1	Ent1	0.90	0.25
Slc38a2	Snat2	0.94	0.70
Slco1a1	Oatp1a1	3.51	0.17
Slco1a4	Oatp1a4	1.26	0.10
Slco1a6	Oatp1a6	1.33	0.08
Slco1c1	Oatp1c1	1.20	0.14
Slco2b1	Oatp2b1	0.94	0.59

**Table 2 pharmaceutics-14-02803-t002:** Significant differential abundant proteins between isolated from 145–150 day old female WT and C9 mice, assessed using TMT-proteomics.

Gene	Protein Name	WT/C9 Fold-Change	Adjusted *p* Value
Atp5mc1	ATP synthase F(0) complex subunit C1	1.63	0.0243
Bcap29	B-cell receptor-associated protein 29	1.51	0.0243
Cdc42bpa	CDC42 binding protein kinase alpha	1.27	0.0446
Gulp1	PTB domain-containing engulfment adapter protein 1	1.56	0.0258
H2ac15	histone H2A type 1-K	1.61	0.0243
Hmgn3	high mobility group nucleosome-binding domain-containing protein 3	1.95	0.0073
Isg15	ubiquitin-like protein ISG15	1.42	0.0243
Nckap1l	nck-associated protein 1-like	0.64	0.0292
Nono	non-POU domain-containing octamer-binding protein	1.41	0.0243
Rgs10	regulator of G-protein signaling 10	0.44	0.0243
Smco4	single-pass membrane and coiled-coil domain-containing protein 4	1.79	0.0243
Tmpo.1	lamina-associated polypeptide 2	1.55	0.0355
UPF0729	UPF0729 protein C18orf32 homolog	1.70	0.0355

## Data Availability

Data is contained within the article or Appendix A.

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
