# Peer review of "Altered Blood–Brain Barrier Dynamics in the C9orf72 Hexanucleotide Repeat Expansion Mouse Model of Amyotrophic Lateral Sclerosis"

_pharmaceutics, 2022, doi:10.3390/pharmaceutics14122803_

Round 1
Reviewer 1 Report
Comments and Suggestions for Authors
The current studies titled “Altered Blood-Brain Barrier Dynamics in the C9orf72 Hexanu-1 cleotide Repeat Expansion Mouse Model of Amyotrophic Lat-2 eral Sclerosis” by Yijun Pan, and associates worked on altered BBB dynamics in different mouse model of lateral Sclerosis. The article is well designed and written with professional approach and data presented is very precise, suitable and clear. However, few points require to address:
Comments 1: In Figure 1, how authors concluded that brain-to-perfusate ratio of probe molecules permeation govern via different diffusion mechanism.
Comments 2. It is advised to give a pictorial presentation of BBB outline expressing brain perfusion of probe molecules.
Comment 3. Conclusion part is too short and suggested to add more insight for better visibility of the readers.
Author Response
Please see the attachment v2.

Reviewer 2 Report
Dear Authors,
Thank you for opportunity reading manuscript entitled "Altered Blood-Brain Barrier Dynamics in the C9orf72 Hexanu 1 cleotide Repeat Expansion Mouse Model of Amyotrophic Lateral Sclerosis" by Yijun Pan et al.
This is very interesting paper. Study is well designed. Introduction provide important information, material and methods are well described.
Results are interesting. The conclusions follow from the entire research work.
This manuscript is excellent. It's really hard to make any improvements.
I have only minor comments, suggestions : Tables are ok, but I suggest to spell out the abbreviations like : TMT, ATP, WT and so on.
In my opinion content is very interesting and I think that this interesting paper should be accepted for publication.
Author Response
Thank you for opportunity reading manuscript entitled "Altered Blood-Brain Barrier Dynamics in the C9orf72 Hexanu 1 cleotide Repeat Expansion Mouse Model of Amyotrophic Lateral Sclerosis" by Yijun Pan et al. This is very interesting paper. Study is well designed. Introduction provide important information, material and methods are well described. Results are interesting. The conclusions follow from the entire research work. This manuscript is excellent. It's really hard to make any improvements.
We thank the reviewer for his/her time reading through our manuscript and appreciated the compliment.
I have only minor comments, suggestions: Tables are ok, but I suggest to spell out the abbreviations like: TMT, ATP, WT and so on.
We agree with the reviewer and have minimise the use of abbreviation where possible, e.g. CSF is now replaced by cerebrospinal fluid. In the case when abbreviation is used, we ensured that they have been spell out in full for the first time, e.g. TMT on page 7, and ATP and WT on page 3.
In my opinion content is very interesting and I think that this interesting paper should be accepted for publication.